# Aldo-Keto Reductase 1C3 Mediates Chemotherapy Resistance in Esophageal Adenocarcinoma via ROS Detoxification

**DOI:** 10.3390/cancers13102403

**Published:** 2021-05-16

**Authors:** Chenghui Zhou, Zhefang Wang, Jiahui Li, Xiaolin Wu, Ningbo Fan, Dai Li, Fanyu Liu, Patrick S. Plum, Sascha Hoppe, Axel M. Hillmer, Alexandar Quaas, Florian Gebauer, Seung-Hun Chon, Christiane J. Bruns, Yue Zhao

**Affiliations:** 1Department of General, Visceral, Cancer and Transplantation Surgery, University Hospital Cologne, 50937 Cologne, Germany; chenghui.zhou@uk-koeln.de (C.Z.); zhefang.wang@zju.edu.cn (Z.W.); jiahuiliss@163.com (J.L.); xiaolin.wu@uk-koeln.de (X.W.); ningbo.fan@uk-koeln.de (N.F.); lidaichhosp@163.com (D.L.); fanyu.liu@student.uni-tuebingen.de (F.L.); patrick.plum@uk-koeln.de (P.S.P.); florian.gebauer@uk-koeln.de (F.G.); seung-hun.chon@uk-koeln.de (S.-H.C.); 2Department of Plastic and Reconstructive Surgery, Second Affiliated Hospital, School of Medicine, Zhejiang University, Hangzhou 310058, China; 3Department of Anesthesiology, Changhai Hospital, Naval Medical University, Shanghai 200433, China; 4Interfaculty Institute for Cell Biology, University of Tübingen, Auf der Morgenstelle 15, 72076 Tübingen, Germany; 5Faculty of Medicine and University Hospital Cologne, Institute of Pathology, University of Cologne, 50937 Cologne, Germany; sascha.hoppe@uk-koeln.de (S.H.); axel.hillmer@uni-koeln.de (A.M.H.); alexander.quaas@uk-koeln.de (A.Q.)

**Keywords:** esophageal adenocarcinoma, AKR1C3, chemotherapy resistance, AKT signaling, ROS regulation, prognosis

## Abstract

**Simple Summary:**

The multidrug resistance of EAC is one of the major obstacles to chemotherapeutic efficiency. Our study aims to explore the molecular mechanism of AKR1C3 as a novel therapeutic target to overcome chemotherapy resistance for EAC patients. We demonstrate that AKR1C3 renders chemotherapy resistance through controlling cellular ROS levels via AKT signaling in EAC cells. Modulation of intracellular GSH levels by AKR1C3 could scavenge the intracellular ROS, thus regulating apoptosis. Targeting AKR1C3 may represent a novel strategy to sensitize EAC cells to conventional chemotherapy treatment and benefit the overall survival of patients diagnosed with EAC.

**Abstract:**

Esophageal adenocarcinoma (EAC) is one of the most lethal malignancies, and limits promising treatments. AKR1C3 represents a therapeutic target to combat the resistance in many cancers. However, the molecular mechanism of AKR1C3 in the chemotherapy resistance of EAC is still unclear. We found that the mRNA level of AKR1C3 was higher in EAC tumor tissues, and that high AKR1C3 expression might be associated with poor overall survival of EAC patients. AKR1C3 overexpression decreased cell death induced by chemotherapeutics, while knockdown of AKR1C3 attenuated the effect. Furthermore, we found AKR1C3 was inversely correlated with ROS production. Antioxidant NAC rescued chemotherapy-induced apoptosis in AKR1C3 knockdown cells, while the GSH biosynthesis inhibitor BSO reversed a protective effect of AKR1C3 against chemotherapy. AKT phosphorylation was regulated by AKR1C3 and might be responsible for eliminating over-produced ROS in EAC cells. Intracellular GSH levels were modulated by AKR1C3 and the inhibition of AKT could reduce GSH level in EAC cells. Here, we reported for the first time that AKR1C3 renders chemotherapy resistance through controlling ROS levels via AKT signaling in EAC cells. Targeting AKR1C3 may represent a novel strategy to sensitize EAC cells to conventional chemotherapy.

## 1. Introduction

Esophageal cancer (EC) is the 7th most commonly diagnosed cancer and the 6th leading cause of cancer-related death worldwide [1]. Esophageal adenocarcinoma (EAC) and squamous cell carcinoma (ESCC) are two major histological subtypes of esophageal cancer. ESCC is the predominant subtype globally, while EAC is the most common in Europe and other Western industrialized nations. It is well recognized that the clinical characteristics and molecular profiles of EAC and ESCC are quite different, despite some shared characteristics, which leads to differential response to clinical treatments [2,3]. Thus, EAC and ESCC should not be considered as one disease but investigated separately. The incidence of EAC has increased sharply in many Western populations during the last four decades [4,5]. Although the treatments for EAC have been improved due to the implementation of modern multimodal treatment concepts such as neoadjuvant chemoradiation or perioperative chemotherapy [6,7], the outcome of EAC is still unfavorable, with a 5-year overall survival rate of about 18% [8].

Aldo-keto reductases (AKRs) are a superfamily of NAD(P)(H)-dependent oxidoreductases that primarily reduce aldehydes and ketones to primary and secondary alcohols, respectively [9]. AKRs play a central role in the metabolism of reactive aldehydes, carcinogens and chemotherapeutic drugs, which could lead to either their bioactivation or detoxication [10]. AKR1C1, AKR1C2, AKR1C3 and AKR1C4 are the members of the aldo-keto reductase 1C (AKR1C) subfamily, which are responsible for the clearance of different xenobiotics, and thus result in resistance to drug treatment [11]. Aldo-keto-reductase 1C3 (AKR1C3), as a member of the AKR superfamily, possesses NADPH-dependent 3-keto-, 17-keto- and 20-ketosteroid reductase activities [12]. In recent years, AKR1C3 has been implicated in the progression of prostate cancer, breast cancer, colon cancer and oropharyngeal squamous cell carcinoma, and contributes to therapeutic resistance in these types of cancers [13,14,15]. However, the functional role of AKR1C3 in EAC remains largely unknown.

One central malignant phenotype of cancer cells is their ability to resist chemotherapy, which is also the main obstacle to effective cancer therapy. A growing number of studies have indicated that the activation of the AKT signaling pathway was implicated in chemotherapeutic resistance in several types of cancers [16,17,18]. AKT, also known as protein kinase B (PKB), is a serine/threonine-specific protein kinase and was initially discovered as a proto-oncogene [19]. Indeed, numerous in vivo and in vitro studies pointed out that AKT kinase is involved in several critical cellular functions of human cancers, including proliferation, migration, survival and apoptosis [20,21]. The regulation of AKT or AKT-related signaling pathways has been the subject of multiple research efforts [21]. For example, the activation of AKT signaling promotes drug resistance such as to cisplatin and paclitaxel in osteosarcoma and ovarian cancer cells, while the inhibition of AKT signaling promotes cell sensitivity to chemotherapy drugs [16,17,18].

Chemotherapeutic agents such as cisplatin, oxaliplatin, 5-fluorouracil and paclitaxel have been identified to eliminate malignant cells in vitro and in vivo by triggering apoptosis [22,23]. However, de-regulated apoptotic signaling allows cancer cells to escape this program, resulting in tumor survival and therapeutic resistance [24]. The cellular redox homeostasis is a balance between oxidation and reduction systems. A large variety of anti-cancer drugs kill cancer cells and overcome drug resistance by disrupting the redox homeostasis of cancer cells [25]. Anti-cancer drugs can induce apoptosis via direct or indirect intracellular reactive oxygen species (ROS) generation in cancer cells [26]. A moderate ROS level is essential for tumor initiation, and plays an important role in cellular signaling pathways that regulate the cell cycle, progression, migration and cell survival [26]. However, excess intracellular ROS may overwhelm the cell’s antioxidant capacity and induce apoptosis [27]. The main amounts of cellular ROS are produced by mitochondria and NADPH oxidases (NOXs) [28]. In order to avoid ROS overproduction, cells use antioxidant molecules and enzymes such as glutathione (GSH) and catalase to eliminate ROS [29].

AKR1C3 has already been implicated in therapeutic resistance via mediating intracellular ROS levels in several types of cancer, such as prostate cancer, ESCC, choriocarcinoma and leukemias [30,31,32,33]. AKR1C3 silencing reverses methotrexate (MTX) resistance in choriocarcinoma cells by increasing ROS levels [33]. In this study, we aimed to investigate the role of AKR1C3 in EAC therapy resistance. Although the interaction between AKR1C3 and chemotherapy resistance has already been indicated in many cancers, it is still unclear in EAC. Based on the above, we make an effort to detect whether AKR1C3 and chemotherapy resistance have a similar interconnection in EAC and whether they might achieve it through an ROS-dependent regulation.

## 2. Materials and Methods

### 2.1. Public Databases

The public database GEO (https://www.ncbi.nlm.nih.gov/geoprofiles/, accessed on 5 December 2020) and TCGA (http://cancergenome.nih.gov/, accessed on 5 December 202) were used for the analysis of AKR1C3 expression in EAC. Gene set enrichment analysis was performed via GSEA software (Version: 4.1.0). Survival analysis was performed using the Kaplan–Meier method and the difference was tested with the log-rank test. A *p*-value < 0.05 was considered statistically significant.

### 2.2. Antibodies and Reagents

Mouse monoclonal anti-α-tubulin (WB, 1:1000, Cell Signaling, 3873, Danvers, Frankfurt, Germany), mouse monoclonal anti-AKR1C3 (WB, 1:1000, R&D Systems, MAB7678, Minneapolis, MN, USA), rabbit monoclonal anti-CXCR4 (WB, 1:1000, Abcam, ab124824, Cambridge, UK), rabbit polyclonal anti-ZEB1(H-102) (WB, 1:500, Santa Cruz, sc-25388, Dallas, TX, USA), rabbit polyclonal anti-SNAIL1 (H-130) (WB, 1:500, Santa Cruz, sc-28199), rabbit monoclonal anti-NRF2 (WB, 1:1000, Cell Signaling, 12721), HRP-conjugated secondary antibody (WB, 1:10,000, Invitrogen, 31430 and 31460), rabbit monoclonal anti-phospho-AKT (Ser473) (WB, 1:1000, Cell Signaling, 4058), rabbit monoclonal anti-phospho-AKT (T308) (WB, 1:1000, Cell Signaling, 13038), rabbit polyclonal anti-AKT (WB, 1:1000, Cell Signaling, 9272), AKT inhibitor (MedChemExpress, HY-10355, Princeton, NJ, USA), IgG control (Cell Signaling, 2729), protein A-Dynabeads (Invitrogen, Carlsbad, CA, USA), N-acetyl-L-cysteine (NAC, Sigma-Aldrich, A9165, Darmstadt, Germany) and L-buthionine-(S,R)-sulfoximine (BSO, MedChemExpress, HY-106376A) were purchased from the indicated manufacturers. University Hospital Cologne supplied chemotherapeutic agents including cisplatin (NeoCorp, Hexal AG, Holzkirchen, Germany), oxaliplatin (Accord HealthCare, München, Germany), paclitaxel (NeoTaxan, Hexal AG) and 5-fluorouracil (5-FU) (Accord HealthCare).

### 2.3. Cell Lines and Clinical Tissues

The human esophageal adenocarcinoma cell lines SKGT-4, FLO-1 and OACP4C were kindly provided by the Laboratory of Genomic Pathology at the Institute of Pathology of University of Cologne (Cologne, Germany), while OE33 was obtained from the Sigma Cell Line Bank (Sigma, 96070808). Cell lines were maintained in RPMI1640 medium (Life technology, Carlsbad, CA, USA) with 10% fetal bovine serum (FBS) (Invitrogen, Carlsbad, CA, USA), penicillin and streptomycin (100 U/mL penicillin + 0.1 mg/mL streptomycin) (PAN Biotech, Aidenbach, Germany) in a humidified atmosphere of 5% CO_2_ at 37 °C. HEK293T cells (Sigma, 12022001) were maintained in DMEM high-glucose medium (Invitrogen) with 10% FBS (Invitrogen, Carlsbad, CA, USA), 2 mM L-Glutamine (Invitrogen), penicillin and streptomycin (100 U/mL penicillin + 0.1 mg/mL streptomycin) in a humidified atmosphere of 5% CO_2_ at 37 °C. All cell lines were checked for mycoplasma-contamination-free culture. Twelve pairs of EAC tissues and adjacent normal tissues were collected for Western blot from the Department of General, visceral, tumor, and transplant surgery of University Hospital of Cologne under the approval of BIOMASOTA (approved by the Ethics Committee of the University of Cologne, ID: 13-091).

### 2.4. Cell Proliferation Assay

EAC cells were seeded into 24-well plates overnight at a density of 1–3 × 10^3^ cells/well and then cultured for 0–8 days. Cells were fixed with 4% paraformaldehyde (PFA) at indicated time points at room temperature (RT) for 10 min, then stained with 0.05% crystal violet (Sigma-Aldrich) at RT for 10 min. After a wash with distilled tap water five times and air drying, 10% acetic acid was added to dissolve the stain, which was subsequently measured with a plate reader (BMG LABTECH, Ortenberg, Germany) by absorbance at 595 nm.

### 2.5. Colony-Forming Assay

EAC cells (5 × 10^2^–1 × 10^3^ cells/well) were seeded in 6-well plates with full RPMI 1640 medium at 37 °C with 5% CO_2_. After being conventionally cultured for two weeks, the cells were fixed and stained with freshly prepared 0.5% crystal violet (Sigma-Aldrich) for 10 min at RT. After a wash with distilled water, the stained colonies with >50 cells were counted by microscopy using 40× magnification (Leica, DMIL, Wetzlar, Germany).

### 2.6. Wound Healing and Migration Assay

Cell migration was assessed by the ability of cells to migrate into a cell-free area. Briefly, 1 × 10^5^ SKGT-4 or OE33 cells were plated in growth medium on 24-well plates. When the cells reached confluence overnight, the monolayers were then artificially wounded by scratching with a 200 µL plastic pipette tip to get a linearly scratched zone. After carefully washing with phosphate-buffered saline (PBS), the SKGT-4 and OE33 cells were incubated in FBS-free medium for 12 h and 24 h, respectively, and observed under a microscope (Leica, DMIL). Images were captured immediately after scratching as a reference point, and were captured again at the end point as indicated. The images were captured using a phase-contrast microscope (Leica, DMIL). The wound closure area measurement was performed by ImageJ. The wound closure rates were estimated as the ratio of the closed wound area relative to the initially wounded area.

### 2.7. Western Blot

Twenty microgram protein samples were electrophoresed on a 7.5–15% gradient SDS-PAGE gel (Tris-Glycine, self-made) and transferred to PVDF membrane (MACHEREY-NAGEL, Dueren, Germany) by semi-dry electroblotting (Bio-Rad, Singapore). The membranes were blocked for 1 h in 1× Roti-Block (Carl Roth, Karlsruhe, Germany) at RT and then incubated with specific primary antibodies at 4 °C overnight. Proteins were detected after incubation with HRP-conjugated secondary antibody (Invitrogen, 31430 and 31460) for 1 h at RT and visualized with SuperSignal West Pico PLUS Chemiluminescent Substrate (Thermo Fisher Scientific, Waltham, MA, USA) and detected by ChemoStar ECL Imager (Intas Science Imaging, Göttingen, Germany).

### 2.8. Cell Viability Assay

Cells were seeded into 96-well plates, grown overnight and then treated with serial concentrations of cisplatin, oxaliplatin, 5-FU or paclitaxel for 48–72 h. Cell viability was detected by MTT assay. Briefly, 50 µL of 5 mM MTT (Biomol, Hamburg, Germany) solution was added to each well after the medium was discarded. Then, the plates were incubated at 37 °C for 3 h. Afterwards, the solution was discarded, and the MTT dissolving agent was added. Absorbance at 570 nm was measured with a plate reader. Cells treated without chemotherapy drugs served as a reference point for standardization. Cell viabilities were calculated as the ratios of absorbance of the wells with various concentrations of chemotherapeutic agents relative to the vehicle control. In all cases biological triplicates were performed.

### 2.9. Flow Cytometry Analysis

Cells were treated with cisplatin for 48–72 h. For apoptosis analysis, cells were harvested and resuspended in annexin V binding buffer, with annexin V (BioLegend, San Diego, CA, USA) and DAPI staining dye, incubated at RT for 20 min. To detect intracellular ROS production, cells were stained with 20 µM H2DCFDA (Sigma-Aldrich) at RT for 20 min. Then samples were subjected to analysis on a CytoFLEX cytometer. Data acquisition and analysis were performed with FlowJo software (Tree Star, Ashland, QR, USA).

### 2.10. Quantification of GSH

The GSH level of EAC cells was determined using a GSH/GSSG Ratio Detection Kit II (Fluorometric-Green, Abcam 205811) according to the manufacturer’s instructions. Before analysis, cells were harvested and washed with ice-cold PBS. Then, 2 × 10^5^ cells were resuspended in 100 µL of ice-cold PBS/0.5% NP-40. The clear supernatant was collected after centrifugation. Then, we added 1 volume ice cold 100% (*w*/*v*) TCA into 5 volumes of sample and vortexed briefly to mix well. Finally, the samples were transferred into a black 96-well plate after the neutralization by NaHCO_3_. Fluorescence was measured with a plate reader at Ex/Em = 490/520 nm. In all cases, biological triplicates were performed.

### 2.11. Plasmid Constructs

For the expression of AKR1C3, the pLenti-CMV-neo vector was purchased from GenScript (Leiden, The Netherlands), and vector with non-coding scrambled insert was used as control. For RNA interference, shRNA sequences for AKR1C3 and NRF2 were synthesized (Thermo Fisher Scientific) and inserted via AgeI and EcoRI into the Tet-pLKO-puro vector. shRNA target sequences were: non-target control, 5′-AGGTAGTGTAATCGCCTTGTT-3′; shAKR1C3-1, 5′-CTCACTGAAGAAAGCTCAATT-3′; shAKR1C3-2, 5′-CCAGAGGTTCCGAGAAGTAAA-3′; shNRF2, 5′-GCTCCTACTGTGATGTGAAAT-3′. All expression vectors were confirmed by sequencing.

### 2.12. Generating AKR1C3 Overexpressing and Knockdown Cell Lines

Cells stably expressing AKR1C3 or shRNA sequence were created by lentiviral transduction. Briefly, HEK293T cells were co-transfected with transfer vector and packaging vectors (Addgene) using PEI (Sigma-Aldrich) in a mass ratio of 1:3 of DNA/PEI. The medium was changed 24 h later, and the virus was collected and filtered through 0.45 µm syringe filters (VWR, Darmstadt, Germany) at 48 h and 72 h. Virus-containing filtrate was mixed 1:1 with fresh medium, supplemented with 8 μg/mL polybrene, and used to transduce cells. Neomycin or puromycin was added 48–72 h later for AKR1C3 overexpressing or knockdown cells, respectively, and the selective medium was changed every 2 days and maintained for 1 week. shRNA expression was induced with 1 μg/mL doxycycline (Sigma-Aldrich). Knockdown and overexpression efficiency was testified by Western blot.

### 2.13. Chromatin Immunoprecipitation

The procedure for ChIP was performed as previously described [34]. Briefly, chromatin samples were subjected to rabbit monoclonal anti-NRF2 (Cell Signaling, 12721) or normal IgG control (Cell Signaling, 2729) at 4 °C overnight. Then, protein–antibody complexes were precipitated with protein A-Dynabeads (Invitrogen). Immunoprecipitated complexes were washed and eluted with buffer (1% SDS, 0.1 M NaHCO_3_) and then incubated with proteinase K for 4 h at 65 °C on a thermomixer. DNA was purified using the PCR Clean Up Kit (MACHEREY-NAGEL) and subjected to quantitative PCR for AKR1C3 promoter detection. The NRF2 binding site in AKR1C3 was predicted by JASPAR and CiiiDER tools. Then, primers were designed using the NCBI primer designing tool to flank the predicted binding site. The used primers were: AKR1C3-ChIP-for, 5′-ACATCTTTACCCCTAGTGTTCAGT-3′; AKR1C3-ChIP-rev, 5′-AGTTCTTGAGATTTTGACTGGATGC-3′.

### 2.14. Quantitative RT-PCR

TRI reagent (Sigma-Aldrich) was used to extract total RNA from cultured cells. Afterward, the High-Capacity cDNA Reverse Transcription Kit (Applied Biosystems, Thermo Fisher Scientific) was used for cDNA synthesis according to the manufacturer’s instruction. Primers are listed in Appendix A. Relative expression of target mRNAs was determined using Fast SYBR Green Master Mix (Invitrogen) with QuantStudio 7 Flex (Applied Biosystems, Thermo Fisher Scientific) and analyzed using the delta-delta-CT method.

### 2.15. Statistical Analysis

Statistical analysis was done by GraphPad Prism 7. The Kaplan–Meier method was used to calculate the overall survival (OS). Data were presented as mean ± SD. Statistical significance was determined by two-sided unpaired *t*-test. 

## 3. Results

### 3.1. Expression and Characterization of AKR1C3 in EAC

To determine the mRNA expression of AKR1C3 in EAC, several public datasets were downloaded from GEO and TCGA. Data from GSE26886 showed that AKR1C3 is upregulated in Barrett’s esophagus (a precursor lesion of EAC), EAC and ESCC as compared to squamous epithelium (Figure 1A). Consistently, GSE92396 showed a similar result (Figure 1B). Additionally, to demonstrate the protein levels of AKR1C3 in our cohort, 12 pairs of EAC tissues and matched adjacent normal tissues were collected. However, no clear trend was observed in our cohort comparing the expression of AKR1C3 between EAC and adjacent normal tissues (Figure 1C). Survival analysis from TCGA revealed that higher expression of AKR1C3 might be associated with poor overall survival of EAC patients, although the data were not statistically significant (Figure 1D).

### 3.2. AKR1C3 Promotes Proliferation, Colony Formation and Migration of EAC Cells

To understand the function of AKR1C3 in EAC cells, we modified AKR1C3 expression in EAC cells through short-hairpin RNA (shRNA) knockdown and overexpression. Four EAC cell lines were selected (Appendix A). AKR1C3 knockdown was established in SKGT-4 and OACP4C cells, which have relatively high endogenous AKR1C3 expression, while AKR1C3 overexpression was established in OE33 and FLO-1 cells, which have relatively low endogenous AKR1C3 expression (Figure 2A and Appendix A). The expression level of AKR1C3 was confirmed by PCR and Western blot. In the knockdown setup, the best overall silencing efficiency was achieved by shAKR1C3-2, which was chosen for further study. As compared with the control cells, AKR1C3 knockdown cells showed a lower rate of cell proliferation and formed fewer colonies in the colony formation assay. In contrast, AKR1C3 overexpressing cells showed an increased rate of cell proliferation and higher clonogenic ability (Figure 2B,C and Appendix A). Furthermore, we investigated the potential role of AKR1C3 in modulating the migration ability of EAC cells using a wound healing assay. The results showed that AKR1C3 knockdown cells had markedly reduced migratory ability as compared to the control groups, whereas AKR1C3 overexpressing cells showed a faster wound closure rate (Figure 2D and Appendix A). Besides, AKR1C3 knockdown cells showed a decreased expression of metastatic marker C-X-C motif chemokine receptor 4 (CXCR4) and epithelial–mesenchymal transition (EMT)-associated factors zinc finger E-box-binding homeobox 1 (ZEB-1) and snail family transcriptional repressor 1 (SNAIL1) at the protein level. (Figure 2E and Appendix A). However, the expression level of these markers did not significantly increase in AKR1C3 overexpressing cells of OE33 (Figure 2E). Instead, an increase was observed in AKR1C3 overexpressing cells of FLO-1 (Appendix A). Taken together, these results suggest that AKR1C3 promotes the proliferation, colony formation and migration of EAC cells in vitro.

### 3.3. AKR1C3 Renders Chemotherapy Resistance to EAC Cells

An apoptosis assay analyzed with flow cytometry and MTT assay was carried out to determine the impact of AKR1C3 on chemotherapy response in EAC cells *in vitro*. As compared with the control cells, AKR1C3 knockdown cells showed more apoptosis upon cisplatin treatment (Figure 3A; Appendix A), whereas AKR1C3 overexpressing cells showed less apoptosis as compared with control cells (Figure 3B; Appendix A). To identify a more general anti-apoptotic role of AKR1C3 in the context of EAC chemotherapy, oxaliplatin, 5-FU or paclitaxel were also applied, as those agents are used within the common chemotherapeutic regimens. Consistent with cisplatin, high levels of AKR1C3 showed protective function against oxaliplatin, 5-FU or paclitaxel induced apoptosis as well (Figure 3C,D and Appendix A). Additionally, GSEA analysis from the TCGA dataset showed that drug-metabolism-related enzymes were enriched in the AKR1C3-high group of EAC (Appendix A), including alcohol dehydrogenase 4 (ADH4), UDP glucuronosyltransferase family 1 member A6 (UGT1A6) and alcohol dehydrogenase 6 (ADH6). These results indicate that AKR1C3 renders chemotherapeutic resistance in EAC cells.

### 3.4. AKR1C3 Mediates Chemo-Resistance through Regulating Redox Homeostasis

Cytotoxic drugs kill cancer cells by increasing intracellular ROS, which subsequently induces apoptosis or necrosis [35]. To further clarify the underlying mechanisms of AKR1C3-mediated chemo-resistance, intracellular ROS levels were determined by flow cytometry. ROS levels were significantly higher in AKR1C3 knockdown cells than in control cells. Consistently, AKR1C3 overexpressing cells showed decreased ROS levels as compared to control cells (Figure 4A; Appendix A). To further validate the role of AKR1C3 in the regulation of ROS, hydrogen peroxide (H_2_O_2_) was applied to induce ROS generation. AKR1C3 knockdown cells showed more apoptosis upon H_2_O_2_ treatment, whereas AKR1C3 overexpressing cells showed less apoptosis compared to control cells (Figure 4B; Appendix A). To confirm that AKR1C3-mediated chemo-resistance is indeed due to its regulation of redox balance, knockdown cells were exposed to chemotherapeutic drugs in the presence of the antioxidant NFF-acetyl-l-cysteine (NAC), while overexpressing cells were treated with chemotherapy in combination with the GSH biosynthesis inhibitor L-buthionine-S,R-sulfoximine (BSO). The results showed that NAC rescued chemotherapy-induced apoptosis in AKR1C3 knockdown cells, while BSO reversed the protective effect of AKR1C3 overexpression against chemotherapy in EAC cells (Figure 4C,D and Appendix A). To further link AKR1C3 to redox balance maintenance, the regulatory role of NRF2 on AKR1C3 was determined. ChIP assay showed enrichment of NRF2 in the promoter region of AKR1C3 in SKGT-4 (Figure 4E,F). Knockdown of NRF2 in SKGT4 significantly decreased the expression of AKR1C3 at both mRNA and protein level (Figure 4G–I). In summary, our data suggest that AKR1C3 alleviates oxidative stress and confers chemo-resistance to EAC cells.

### 3.5. AKT Phosphorylation Is Regulated by AKR1C3 and Is Responsible for ROS Alleviation in EAC Cells

To explore the mechanism of AKR1C3’s regulation of ROS-mediated apoptosis, the AKT pathway was evaluated. Our results showed that the phosphorylation level of AKT was decreased in AKR1C3 knockdown cells and augmented in AKR1C3 overexpressing cells (Figure 5A; Appendix A). These data support the reported role of AKT in promoting cell survival and inhibiting apoptosis [36]. To confirm the essential role of AKT in AKR1C3-mediated chemo-resistance in EAC cells, AKT inhibitor VIII was applied. The results showed that AKT inhibitor VIII could increase the level of ROS and diminish the protective effect of AKR1C3 against chemotherapy in AKR1C3 overexpressing cells (Figure 5B,C; Appendix A). Importantly, we observed that the level of GSH decreased in AKR1C3 knockdown cells and increased in AKR1C3 overexpressing cells (Figure 5D,E; Appendix A). In addition, GSEA analysis of several datasets showed that the glutathione metabolism signature is positively associated with AKR1C3 expression (Figure 5F). To further clarify whether GSH could be regulated by AKT signaling, EAC cells were treated with AKT inhibitor VIII, and the result showed that GSH was decreased (Figure 5D; Appendix A). In conclusion, these results indicate that AKR1C3/AKT may effectively regulate the synthesis of GSH, which could directly eliminate the intracellular ROS and decrease cell apoptosis upon chemotherapy in EAC.

## 4. Discussion

Although advances in combination chemotherapy and/or radiotherapy have prolonged the overall survival of EAC patients, the high rate of resistance to conventional chemotherapy is still the main obstacle to the effective therapy of EAC [37,38]. AKR1C3, as a key member of the AKR1Cs subfamily, has been identified as a potential novel therapeutic target in multiple types of cancer [13,39,40]. Recently, AKR1C3 has been reported to be upregulated in many human tumors and identified as a prognostic marker in various cancers, including breast cancer, prostate cancer and colon cancer [14,15,41].

The expression of AKR1C3 is elevated in EAC as compared to normal esophagus according to several public datasets. The inconsistent expression of AKR1C3 in our cohort might be explained by the possibility that some of the esophageal biopsies are Barrett’s esophagus in our cohort. Barrett’s esophagus is considered as the main precursor of EAC, and the expression of AKR1C3 in Barrett’s esophagus is also elevated as compared to normal esophageal squamous epithelium. We found that AKR1C3 promoted proliferation, colony formation and migration in our four EAC cell lines, indicating the crucial role of AKR1C3 in EAC. Here, we report that AKR1C3 positively regulated the phosphorylation of AKT. AKT is a key component in multiple signaling pathways and participates in multiple cellular processes [36]. Numerous studies have demonstrated that AKT mediates tumor progression mainly through inhibiting apoptosis [36,42]. AKT suppresses apoptosis mainly by regulating many downstream effectors such as GSK and Bcl-2 family proteins [36]. The phosphorylation of AKT promotes drug resistance by protecting cells from apoptosis [42]. Furthermore, several studies have shown that the overexpression and/or activation of AKT results in resistance to cisplatin in several types of cancer such as ovarian cancer, cervical cancer and gynecological carcinoma [18,43,44]. Based on the present work performed in EAC cells, we demonstrated that AKR1C3 resulted in EAC cells with greater resistance to chemotherapeutic drugs via the activation of AKT.

Interestingly, we found that an upregulated expression of AKR1C3 in EAC cells could decrease the intracellular ROS levels. AKR1C3 possesses oxidoreductase activity and aldo-keto reductase (NADP/H) activity, and participates in the regulation of cell redox homeostasis [12]. ROS have a dual role in cancer. Moderate ROS are required for essential cellular functions such as gene expression, while excessive ROS diminish cellular antioxidant capacity, thereby damaging cellular structures and leading to apoptosis directly [45]. It is widely accepted that most chemotherapeutics elevate intracellular levels of ROS, thereby altering the redox homeostasis of cancer cells [46]. Several studies have suggested that cisplatin increases the generation of intracellular ROS, while excessive ROS levels could cause oxidative DNA damages and accelerate cell death [45,47,48]. Chueca et al. reported that excessive ROS could induce apoptosis in EAC cells [49]. Under the selective pressure induced by chemotherapeutic drugs, cancer cells would evolve an antioxidant system to protect themselves against oxidative stress via neutralizing the over-produced intracellular ROS [50,51]. Although the connection of AKR1C3 with ROS regulation has already been indicated in prostate cancer and ESCC, these studies mainly focused on radiation resistance, and the underlying molecular mechanism of the regulatory role of AKR1C3 with ROS is not well investigated [31,32]. Surprisingly, we observed that increased expression of AKR1C3 protected EAC cells from apoptosis by scavenging the over-produced ROS. By adding NAC in AKR1C3 knockdown cells and adding BSO in overexpressing cells, we demonstrated that ROS regulation is the main mechanism of AKR1C3-mediated chemo-resistance in EAC. We also report that AKR1C3 positively regulates GSH to enhance cellular antioxidant defense, which appears to contribute significantly to the protection against chemotherapy-induced toxicity. GSH, a tripeptide composed of glutamate, is an important antioxidant in cells [52]. As the key antioxidant guardian, it is generally recognized that GSH could neutralize intracellular ROS and decrease oxidative stress directly [53]. Our results showed that the ratio of apoptotic cells decreased after the addition of NAC, which is a precursor of glutathione (GSH). Inhibition of the biosynthesis of GSH reversed the protective effect of AKR1C3 overexpression against cisplatin by exacerbating oxidative stress in EAC cells. Taken together, our results indicate that AKR1C3 inhibits apoptotic cell death by alleviating oxidative stress in EAC through the regulation of GSH.

In the present study, we demonstrated that AKR1C3 scavenges the intracellular ROS levels in EAC cells via AKT/GSH signaling. Owing to the selective pressure induced by high ROS levels, a series of concomitant changes will occur in cancer cells [45]. For example, recent evidence suggests that excessive oxidative stress activates AKT signaling by inhibiting the activity of phosphatase and tensin homolog (PTEN) and regulating protein tyrosine phosphatases (PTPs) [54]. High ROS levels not only modulate PTEN oxidation directly but also facilitate PTEN ubiquitylation and degradation by inducing post-translational modification [55]. PTPs could dephosphorylate insulin receptor substrate protein 1 (IRS-1) in response to insulin-induced ROS and activate PI3K/AKT pathway subsequently, which plays a significant role in chemotherapy resistance [56]. Activation of the PI3K/AKT signaling pathway has been shown to be essential in modulating glutathione metabolism [57]. Besides, the activation of AKT is crucial in the regulation of GSH synthesis [57,58,59]. Kim et al. revealed that the AKT/p70S6K pathway was capable of manipulating the GSH biosynthesis activated by insulin [60]. Wu et al. reported that the activation of AKT/NRF2 could upregulate the synthesis of GSH in human lung epithelial cells [61]. Our study found that AKR1C3 is a direct target of NRF2, which is an essential regulator of redox balance. Besides, both AKR1C3 knockdown and AKT inhibition reduced GSH levels significantly, which indicates the critical roles of AKT in response to alleviating oxidative stress by increasing antioxidant responses. At the same time, the overexpression of AKR1C3 increased GSH levels in EAC cells. Supportively, public data analysis using the GSEA method reveals that GSH metabolism signature is positively associated with AKR1C3 expression. Therefore, AKR1C3 activates AKT and increases GSH chronologically, through which process it scavenges the overproduced ROS; thus, EAC cells may be protected from apoptosis.

It is also possible that AKR1C3 mediates chemotherapy resistance through the function of drug metabolism. Li et al. found that inhibiting the 11-ketoprostaglandin reductase activity of AKR1C3 could enhance the radiation sensitivity in ESCC [32]. In addition, Matsunaga et al. reported that AKR1C3 showed doxorubicin-reductase activity in gastrointestinal cancer cells [62]. Consistently, GSEA analysis from the TCGA dataset shows drug-metabolism-related enzymes upregulated in the AKR1C3-high group (Appendix A), which may further support the role of AKR1C3 in chemo-resistance. Further investigation is needed for a deeper understanding of the role of AKR1C3 in drug metabolism.

## 5. Conclusions

Our results demonstrate that AKR1C3 might modulate chemotherapy resistance in EAC. Via the AKT signaling pathway, AKR1C3 could accumulate GSH, which can neutralize intracellular ROS levels and decrease oxidative stress, thus finally resulting in the chemo-resistance in EAC (Figure 6). Therefore, we propose that AKR1C3 may act as a potential molecular marker to predict chemotherapy response in EAC cells. A better understanding of the complex interplay between AKR1C3 and redox homeostasis provides more potent therapeutic combinations for EAC to overcome the conventional therapy resistance.

## Figures and Tables

**Figure 1 cancers-13-02403-f001:**
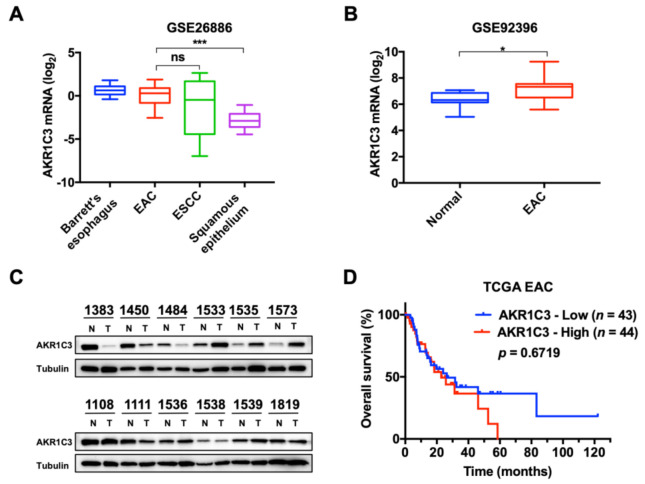
AKR1C3 expression was upregulated in esophageal adenocarcinoma cancer and indicated poor prognosis from public databases. (**A**) Date of GSE26886 showed that AKR1C3 expression level was significantly higher in Barrett’s esophagus, EAC and ESCC than in the corresponding squamous epithelium (BE, *n* = 20; EAC, *n* = 21; ESCC, *n* = 9; Squamous epithelia, *n* = 19). (**B**) Date of GSE92396 also showed that AKR1C3 expression level was significantly higher in EAC than in the corresponding normal esophageal tissue (Normal, *n* = 10; EAC, *n* = 12). (**C**) Western blot results showed the expression of AKR1C3 in 12 pairs of EAC tissues (T) and matched adjacent normal tissues (N). (**D**) Data from TCGA-ESCA was applied for survival analysis and the EAC subgroup was extracted. Kaplan–Meier survival analysis shows that higher AKR1C3 mRNA expression is associated with a trend of poor survival. * *p* < 0.05, *** *p* < 0.001.

**Figure 2 cancers-13-02403-f002:**
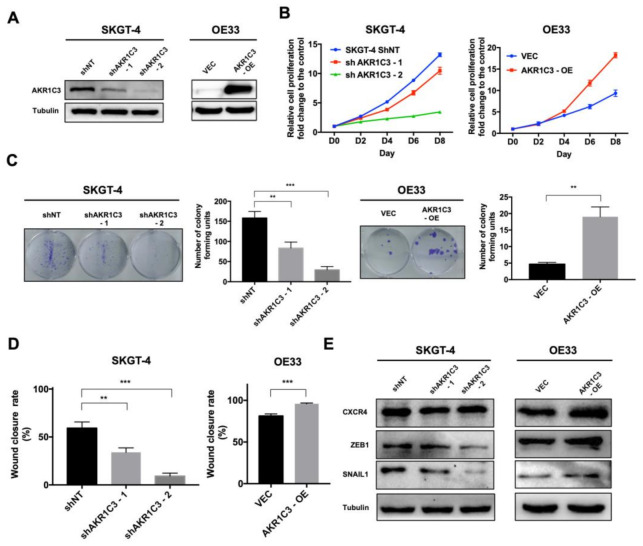
AKR1C3 promotes proliferation, colony formation and migration of EAC cells. (**A**) Validation of stable knockdown of AKR1C3 in SKGT-4 cells and overexpression of AKR1C3 in OE33 cells. (**B**) Proliferation rates, (**C**) colony forming capacity and (**D**) migration capacity of SKGT-4 shAKR1C3 cells, OE33 AKR1C3 overexpressing cells and their respective control cells were examined. (**E**) Western blot results showed that the expression levels of CXCR4, ZEB1 and SNAIL1 were downregulated when AKR1C3 was knocked down in SKGT-4 cells, but these markers did not significantly increase when AKR1C3 was overexpressed in OE33 cells. Data are presented as mean ± SD (*n* = 3). ** *p* < 0.01, *** *p* < 0.001.

**Figure 3 cancers-13-02403-f003:**
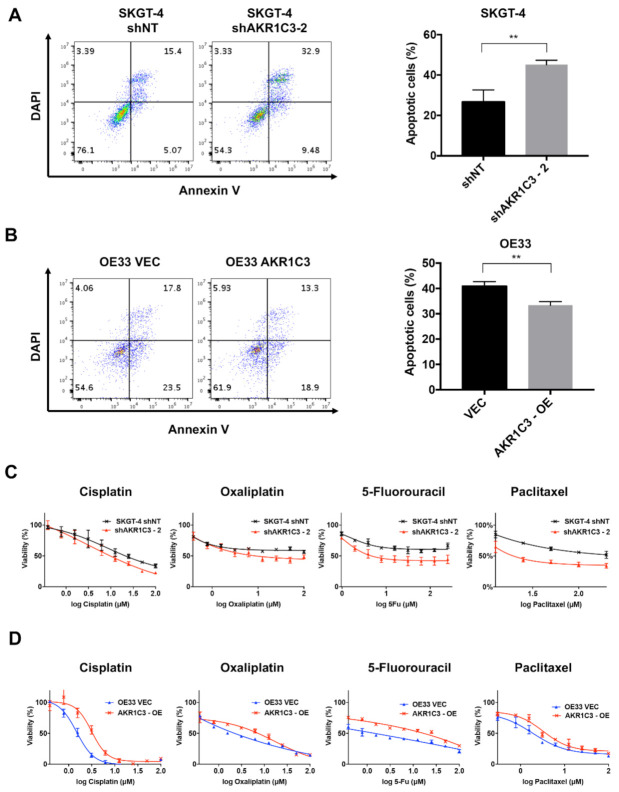
AKR1C3 renders chemotherapy resistance in EAC cells. (**A**,**B**) Cells were treated with cisplatin (20 µM for SKGT-4 and 10 µM for OE33) for the indicated time, and apoptosis was determined by flow cytometry analysis with annexin V/DAPI staining. Representative FACS dot plots are shown on the left. Bar graphs are presented as mean ± SD of three independent experiments. ** *p* < 0.01. (**C**) Cell viability assay was used to determine cell viability after treatment with serial concentrations of cisplatin (0 to 100 µM), oxaliplatin (0 to 100 µM), 5-FU (0 to 250 µM) or paclitaxel (0 to 200 µM) treatment in SKGT-4 for 24 h. (**D**) Cell viability assay was used to determine cell viability after treatment with serial concentrations (0 to 100 µM) of cisplatin, oxaliplatin, 5-FU or paclitaxel in OE33 cells for 24 h.

**Figure 4 cancers-13-02403-f004:**
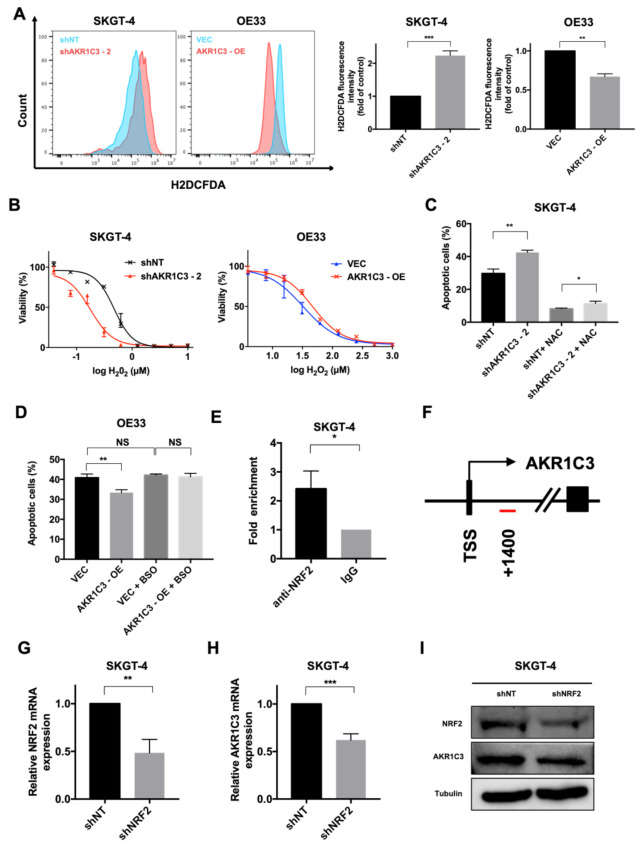
AKR1C3 mediates chemo-resistance through regulating redox homeostasis. (**A**) Intracellular ROS production was measured by flow cytometry using H2DCFDA. Representative FACS histograms are shown on the left. Bar graphs are shown on the right panel. Data are presented as mean ± SD of three independent experiments. (**B**) Cell viability assay was used to determine cell viability after hydrogen peroxide treatment for 24 h in SKGT-4 and OE33 cells. (**C**) SKGT-4 cells were incubated with 4 mM NAC and 10 µM cisplatin for 48 h, followed by the measurement of apoptosis (annexin V/DAPI flow cytometry, bar charts). (**D**) OE33 cells were incubated with 50 µM BSO and 10 µM cisplatin for 48 h, followed by measurement of apoptosis (annexin V/DAPI flow cytometry, bar charts). (**E**,**F**) Chromatin immunoprecipitation assay indicates direct binding of NRF2 to the promoter region of AKR1C3 (around transcription start site (TSS) +1400, indicated by a red bar). (**G**–**I**) NRF2 mRNA levels and protein levels were analyzed after NRF2 knockdown in SKGT-4 cells. * *p* < 0.05, ** *p* < 0.01, *** *p* < 0.001, NS: non-significant (*p* > 0.05).

**Figure 5 cancers-13-02403-f005:**
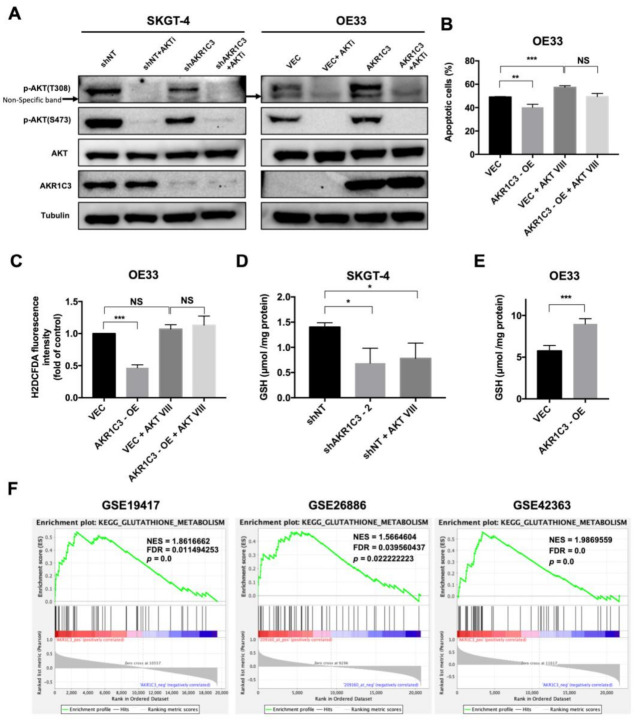
AKT phosphorylation was regulated by AKR1C3 and was responsible for ROS alleviation in EAC. (**A**) Western blot showed that the phosphorylation level of AKT was downregulated in AKR1C3 knockdown cells and upregulated in AKR1C3 overexpressing cells, and the phosphorylation of AKT expression levels was lower in cells treated with 20 µM AKT inhibitor VIII than in cells without inhibitor, but total AKT protein level did not differ among groups. (**B**) OE33 cells were incubated with 20 µM AKT inhibitor VIII and 10 µM cisplatin for 48 h followed by measurement of apoptosis (annexin V/DAPI flow cytometry, bar charts). (**C**) OE33 cells were incubated with 20 µM AKT inhibitor VIII for 24 h followed by measurement of ROS (H2DCFDA dye flow cytometry, bar charts). (**D**,**E**) Data indicate glutathione concentrations normalized to total protein content were decreased in SKGT-4 shAKR1C3-2 cells and increased in OE33 AKR1C3 overexpressing cells, while glutathione concentration was lower in cells treated with 20 µM AKT inhibitor VIII as compared to the control. (**F**) Gene set enrichment analysis shows that glutathione metabolism signature is positively associated with AKR1C3 expression. Data are shown as mean ± SD of triplicate samples. * *p* < 0.05, ** *p* < 0.01, *** *p* < 0.001, NS: non-significant (*p* > 0.05).

**Figure 6 cancers-13-02403-f006:**
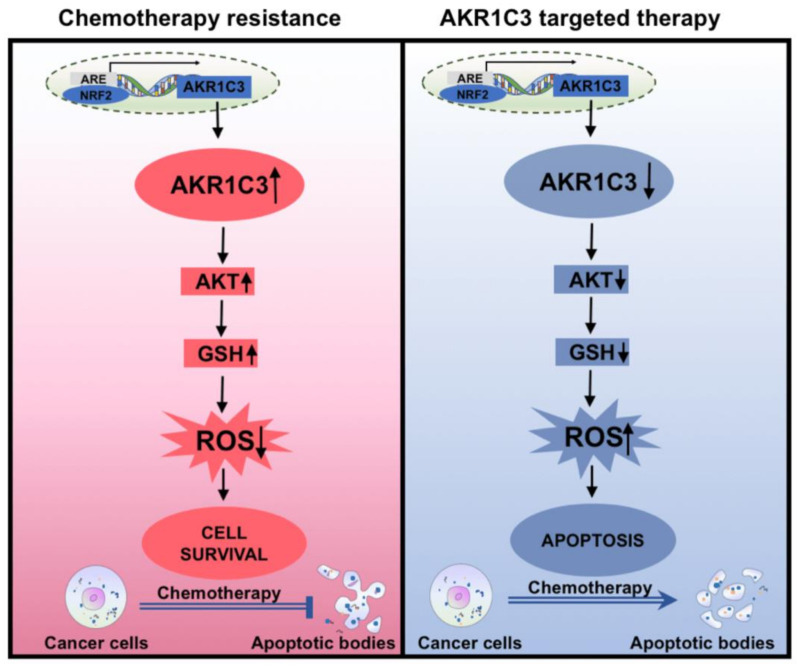
Schematic picture of AKR1C3 regulating intracellular ROS levels and protecting EAC cells from chemotherapy-induced apoptosis.

## Data Availability

Data is contained within the article and Appendix A.

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
