# Peer review of "Aldo-Keto Reductase 1C3 Mediates Chemotherapy Resistance in Esophageal Adenocarcinoma via ROS Detoxification"

_cancers, 2021, doi:10.3390/cancers13102403_

Round 1

Reviewer 1 Report

In the manuscript “Aldo-keto reductase 1C3 mediates chemotherapy resistance in esophageal adenocarcinoma via ROS detoxification” the authors are looking for an additional marker that can help to better predict chemoresistance in the treatment of esophageal cancer. They have chosen as a candidate the enzyme AKR1C3, which is described to give therapeutic resistance to some cancer types. With their experiments, the authors were able to determine which mechanisms are regulated by AKR1C3 in such a way that chemoresistance can develop in esophageal adenocarcinoma cell lines. They were able to show that the presence of AKR1C3 activates the AKT pathway, which increases the level of antioxidant glutathione leading to a decrease in reactive oxygen species. The lack of oxidative stress results in less cell death by apoptosis, a sign of chemoresistance.

The outline of the project, the experiments and the data look very sound. However, the material and methods chapter has to be revised thoroughly, and especially primer info has to be checked.

Additionally, some typing errors and oversights have to be removed. At some points the use of past and present has to be checked.

In detail, there are several issues:

General

Define and unify naming of plasmids, transfected or treated cells, shRNAs, …

Concerning cells, I would say “AKR1C3 overexpressing cells” instead of  “AKR1C3 overexpression cells” or “AKR1C3 overexpressed cells”. 

Throughout the text: Make more clear where mRNA expression and where protein expression is addressed. Applying the nomenclature according to international conventions could be helpful (gene expression: gene name in italic, protein expression: protein name not italic and capital for human proteins).

Introduction

The aims of the study should be explained in a little more detail. Beside, the AKR1C3 part in the introduction should appear directly before the aims.

Line 62:  add an “of” before “about..”

Line 82:  add an “to” before “cisplatin..”

Line 87: I would replace “the” by “a”

Lines 91-93. Rephrase sentence “Anti-cancer therapies…”

Line 96: replace “sources” by “amounts”

Material & Methods

General:

Concerning replicates: please indicate whether biological or technical replicates

Some methods are described thoroughly, others are missing important details. Please add missing information. Unify referencing for material sources.

Chapter 2.1 Antibody and Reagents

Please complete: Tubulin antibody is missing, other antibodies appear in other chapters

Line 128 correct concentrations “1% penicillin and streptomycin” to molarity and units

Line 130:  “..were collected from biobank..” Name of biobank?

Line 135: “4% paraformaldehyde (PFA) incubation” how long at which temperature?

Line 137:  replace “air-dried” by “air drying”

Line 138:  replace “staining and” by “stain which”

Line 141: full medium used or RPMI without FBS?

Line 144: “Colony-forming efficiency was calculated …” How was calculation done?

Line 146: I would change the headline into “Wound healing and migration assay”

Line148: “…, cells were plated 1×105 cells in growth medium…” please rephrase to get a correct sentence

Line 149:  How broad were the scratches? Does the broadness influence the rate of healing?

Line 151-152: How was monitoring done? Software used?

Chapter 2.7: Information on Western blotting for tissue samples is missing

Line 158: 7.5% - 15% gradient SDS-PAGE gel from a company or self made? Tris glycin?

Line 159: transfer to PVDF how?

Lines 162, 215, 216: why are the used antibodies not mentioned in the Antibody and Reagents chapter? In these sections here the dilutions of the antibodies would be of interest.

Line 166: Rephrase headline to “Cell viability assay”

Lines166-171: How was normalization in viability assays done?

Line 167: Which drugs were used? This is found only later in line 173 but should be mentioned already here. Which concentrations were used?

Line 168: MTT assay: from which company?

Lines 170, 187: Plate reader name? Biological or technical replicates?

Line 173: were the cells treated with all 4 drugs at the same time? If not replace “and” with “or”. Which concentrations were used?

Chapter 2.11 and 2.12 should better be arranged differently: One should describe overexpression of AKR1C3, the other the generation of constitutive knockdown cells

Chapter 2.11 How was AKR1C3 overexpression achieved? Stable transfection?

Line 189: Name of AKR1C3 vector is missing.

A blank vector is also used in the experiments: kind; supplier?

Lines 192-194: check primers for NRF2. One is the duplication of an AKR primer, one recognizes VDAC2

Chapter 2.12

Line 197: headline: choose a more suitable title

Maintenance of stably transfected cells is not described

Line 205: why are HEK293T cells described although they are not used in the experiments.

Line 212: replace “cytosol” by “cytosolic”

Line 224: “Quantity” has to be replaced by “Quantitative”

Line 228 Please correct “Relative Expression of target mRNAs was determined…”

Lines 221-223: “PCR for AKR1C3 promoter detection.” The PCR with these primers seems to be not promoter detection, because the primers are both positioned in the cds. In addition, both primers are given as reverse primers. Please check.

Chapter 2.14: It has to be made sure that GAPDH the right reference for mRNA quantification, i.e., not itself regulated with drug treatment.

Line 241: Replace “the” by “a regulation of the”

Line 242: Rephrase the sentence “However, western blot showed … and be more precise in statements.

 Line 245 add “data were” in front of “not statistically…”.

Results

Line 265: I would phrase “…, the best overall silencing efficiency was achieved by shAKR1C3-2…”

Stick to the name “shAKR1C3-2” instead of shAKR1C3” in the remaining text and figures of the manuscript

Line 275: It is stated that  “In addition, AKR1C3 overexpression induced the opposite results (Fig. 2E and Fig. S1K).” This cannot be read without doubt from the figures. Were densiometric measurements performed to verify the statement?

Line 289: Change “in” to “to”

Line 290-292: Rephrase sentence: “To better characterize …” The beginning of the sentence suggest that you already know what you would like to show with your experiments.

Line 298: add “high levels of” before “AKR1C3”

Line 299: Replace “Except that” by “Additionally”

Lines 299-301: It I stated that “GSEA analysis from TCGA dataset showed drug metabolism related enzymes is up-regulated in AKR1C3-high group of EAC (Fig. S2E). Please name some of the enzymes. From the Fig. 2SE and 5F this information cannot be extracted.

Line 309: replace “and” by “or”

Line 315 remove “that”

Line 348: Headline: add “cells” or “cell models” after “EAC”

Lines 350-352: Verify the statement for OE33 results. Band intensity changes cannot be read from the figures without doubt. Especially for p-AKT (T308) it is not clear which band is the one that’s important. Were densiometric measurements performed to verify the statement?

Lines: 360-362: It is stated that “To further clarify whether GSH is regulated

through AKR1C3/AKT axis, EAC cells were treated with AKT inhibitor VIII, the result showed that GSH is decreased (Fig. 5D; Fig. S4D)” Does this experiment really prove that AKR1C3 especially is needed in the AKT pathway?

Discussion

The first sentence starts with a statement that reads like a conclusion: “Our work demonstrates that AKR1C3 could lead to chemotherapy resistance in EAC.”  Thus, I would shift this and the 2 following sentences (inclusive figure) to the conclusion or exchange/merge them with the current conclusion.

Lines of argumentation should be more targeted, because now the discussion reads in some parts redundant.

Lines 391-392: would you have characteristic markers for Barrett’s esophagus cells to check the hypothesis by WB or PCR?

Line 412 correct “(NADP)” to “(NADP/H)”

Line 444: give full name of PTEN

GSEA analyses: Which key enzymes are up-regulated? Give at least some names with function of the enzymes. Figure S2E does not give any valuable information without the details.

Figures

Figure 1:

make clear that A) and B) show RNA expression data

please add sample numbers for the groups in A) and B)

Line 251: add “squamous epithelium“

Line 254: TCGA EAC which dataset?  are these protein or mRNA expression data?,

What means N and T and the numbers in Fig 1C?

The antibody against AKR1C3: is it specific against AKR1C3 and not cross-reacting with other AKR1C isoforms?

Figure 2

Line 283: The sentence “Representative images …” would have to be shifted to C) since a picture for the Migration assays is missing. Anyhow, the picture for SKGT-4 in Fig.2C looks not like a colony forming experiment since the colonies are not nicely distributed. Please check.    

Line 285: commas are missing in the enumeration of genes

Lines 285-287: are statements conform with the pictures in Fig.2E: Please check 

Line 286: “knockdown” has to be replaced by “knocked down”

Figure 2D: at which time points was analysis done? Who is the 100%?

Figure 3

Fig 3A: some labels are incomplete

Fig3C: is concentration unit for cisplatin correct?

Line 305: Treatment of the different cells strains with different cisplatin concentrations why?

Line 308: rename “Cell cytotoxicity assay” into “Cell viability assay”

Figure 4

Fig4A: why is x-axis described by DCFDA” instead of H2DCFDA”?

Fig 4B: SD are missing? replicates performed?

Line 344: please rephrase “(around TSS +1400, indicated by black bar).” because there are more black bars and TSS is not explained

Line 345: add “NRF2” in front of  “mRNA…”

Make changes in supplement accordingly.

Author Response

To reviewer 1#:

Dear Reviewer,

We appreciated all your advice and guidance. It has led us to better understand the strength and weaknesses of our manuscript.

Below is our response and update for your review:

Comment on General part:

  1. Define and unify naming of plasmids, transfected or treated cells, shRNAs, …

Response: We unified the name of plasmids, transfected or treated cells, shRNA in the manuscript. All have been marked in the revision.

  1. Concerning cells, I would say “AKR1C3 overexpressing cells” instead of “AKR1C3 overexpression cells” or “AKR1C3 overexpressed cells”.

Response: We apologize for the writing and grammatical errors, and they have been corrected in the manuscript. We renamed “AKR1C3 overexpression cells” or “AKR1C3 overexpressed cells” by “AKR1C3 overexpressing cells”, as you mentioned in the manuscript.

  1. Throughout the text: Make more clear where mRNA expression and where protein expression is addressed. Applying the nomenclature according to international conventions could be helpful (gene expression: gene name in italic, protein expression: protein name not italic and capital for human proteins).

Response: Thank you so much for your reminder and suggestions. We checked the mRNA expression and protein expression of all figures carefully and apply the nomenclature accordingly to your very valuable guidance.

Comment on Introduction part:

  1. The aims of the study should be explained in a little more detail. Besides, the AKR1C3 part in the introduction should appear directly before the aims.

Response: Thank you for the advice. We have explained the aims of the study more in detail. Considering the overall framework and the structure of the full text, we have added a part of description and a summary of AKR1C3 before the aims of the study. The final version has been presented in the manuscript (Page 2, lines 74-78).

  1. Line 62: add an “of” before “about..”

 Response: it has been revised as suggested accordingly.

  1. Line 82: add an “to” before “cisplatin..”

Response: it has been revised as suggested accordingly.

  1. Line 87: I would replace “the” by “a”

Response: it has been revised as suggested accordingly.

  1. Lines 91-93. Rephrase sentence “Anti-cancer therapies…”

Response: We rephrased the sentence “Anti-cancer therapies based on oxidative damage through direct intracellular reactive oxygen species (ROS) generation, which is an essential step of the apoptotic process.” to “Anti-cancer drugs can induce apoptosis via direct or indirect intracellular reactive oxygen species (ROS) generation in cancer cells.”

  1. Line 96: replace “sources” by “amounts”

Response: it has been revised as suggested in the context.

Comment on Material & Methods part:

  1. Concerning replicates: please indicate whether biological or technical replicates

Response: Experiments were replicated biologically.

  1. Some methods are described thoroughly, others are missing important details. Please add missing information. Unify referencing for material sources.

Response: Regarding the incomplete content of the methods, we have added some more detail and essential information and re-organized the materials in the manuscript according to the comments.

  1. Please complete: Tubulin antibody is missing, other antibodies appear in other chapters

Response:  We apologize for the missing information of Tubulin antibody, it’s now integrated in the revision.

  1. Line 128 correct concentrations “1% penicillin and streptomycin” to molarity and units

Response: We updated the final concentration of the penicillin and streptomycin as “100 U/ml Penicillin + 0.1 mg/ml streptomycin”.

  1. Line 130: “..were collected from biobank..” Name of biobank?

Response: We acquired the samples from Department of General, visceral, tumor, and transplant surgery of University Hospital of Cologne under the approval of BIOMASOTA (approved by the Ethics Committee of the University of Cologne, ID: 13-091), the information has been updated in the revised manuscript.

  1. Line 135: “4% paraformaldehyde (PFA) incubation” how long at which temperature?

Response: Cells were fixed with 4% paraformaldehyde (PFA) at indicated time points at room temperature for 10 min. We added the information within the manuscript.

  1. Line 137: replace “air-dried” by “air drying”

Response: It has been revised as suggested.

  1. Line 138: replace “staining and” by “stain which”

Response: It has been revised as suggested.

  1. Line 141: full medium used or RPMI without FBS?

Response: Full RPMI 1640 medium with 10% FBS was used for colony-forming assay.

  1. Line 144: “Colony-forming efficiency was calculated …” How was calculation done?

Response: We apologize for the improper expression. Actually, we count the colonies with > 50 cells and compare the colony number directly.  

  1. Line 146: I would change the headline into “Wound healing and migration assay”

Response: It has been revised as suggested.

  1. Line148: “…, cells were plated 1×105 cells in growth medium…” please rephrase to get a correct sentence

Response: we rephrase the sentence in the manuscript accordingly.

  1. Line 149: How broad were the scratches? Does the broadness influence the rate of healing?

Response: The monolayers were wounded by scratching with a 200µl plastic pipette tip to get a linearly scratched zone. Therefore, the width of the scratched zones was comparable throughout our experiments. Even though the broadness might influence the rate of healing, we normalized the wounded area. More details were added in the manuscript.

  1. Line 151-152: How was monitoring done? Software used?

Response: Images of EAC cells with scratching were captured for the initially wounded area immediately, which was served as a reference point for standardization. Then the images were captured again for the final wound area at indicated time points by using a phase-contrast microscope. The wound closure area measurement was performed by Java based ImageJ software. The wound closure rates were estimated as the ratio of the closed wound area relative to the initially wounded area.

  1. Chapter 2.7: Information on Western blotting for tissue samples is missing

Response: Information on Western blotting for tissue samples was now updated in the Chapter: 2.3. Cell lines and clinical tissues.

  1. Line 158: 7.5% - 15% gradient SDS-PAGE gel from a company or self-made? Tris glycin?

Response: We applied 7.5% - 15% gradient SDS-PAGE gels (Tris-Glycine), which were self-made in this study.

  1. Line 159: transfer to PVDF how?

Response: We transferred the proteins to PVDF membrane by Semi-dry electroblotting (Bio-Red, Singapore), the relevant information was now updated.

  1. Lines 162, 215, 216: why are the used antibodies not mentioned in the Antibody and Reagents chapter? In these sections here the dilutions of the antibodies would be of interest.

Response: We apologized that we missed some antibodies in the Antibody and Reagents chapter, and we checked all antibodies that we used in this study and completed all the antibody information including the dilutions of the application.

  1. Line 166: Rephrase headline to “Cell viability assay”

Responses: It has been revised as suggested.

  1. Lines166-171: How was normalization in viability assays done?

Response: The information about normalization in viability assays has been added in the manuscript.

  1. Line 167: Which drugs were used? This is found only later in line 173 but should be mentioned already here. Which concentrations were used?

Response: Cisplatin was used for the Flow cytometry analysis, and cisplatin, oxaliplatin, 5-FU or paclitaxel were used for cell viability assay. The concentration also has now been added in the manuscript accordingly.

  1. Line 168: MTT assay: from which company?

Response: MTT was purchased from Biomol, Germany.

  1. Lines 170, 187: Plate reader name? Biological or technical replicates?

Response: The name of the plate reader is BMG LABTECH, which is updated now in the revised manuscript.

  1. Line 173: were the cells treated with all 4 drugs at the same time? If not replace “and” with “or”. Which concentrations were used?

Response:  Sorry for the imprecise description in the manuscript. Those four drugs were not used at the same time, and we rewrote the information accordingly. The detail information of concentration was added in the figure legends.

  1. Chapter 2.11 and 2.12 should better be arranged differently: One should describe overexpression of AKR1C3, the other the generation of constitutive knockdown cells

Response: Chapter 2.11 Plasmid constructs contain both the OE and KD plasmid. And we now clarify the information in Chapter 2.12 to generate AKR1C3 overexpressing and knockdown cell lines.

  1. Chapter 2.11 How was AKR1C3 overexpression achieved? Stable transfection?

Response: We apologized that we missed this part. Thank you for the advice. AKR1C3 overexpression was stably overexpressed by lentiviral transduction, and we updated this into the revision.

  1. Line 189: Name of AKR1C3 vector is missing.

Response: We added the name of AKR1C3 vector: pLenti-CMV-neo vector.

  1. A blank vector is also used in the experiments: kind; supplier?

Response: The pLenti-CMV-neo vector was purchased from Genscript, using scramble vector as control.

  1. Lines 192-194: check primers for NRF2. One is the duplication of an AKR primer, one recognizes VDAC2

Response: We deeply apologize for this mistake; we now corrected the primers information for NRF2.

  1. Line 197: headline: choose a more suitable title

Response: We replaced the title: “Lentiviral transduction” with “Generating AKR1C3 overexpressing and knockdown cell lines”.

  1. Maintenance of stably transfected cells is not described

Response: We apologize that the information is not made clearly. After selection with antibiotics, cells were stably transfected with OE or KD plasmids, no further maintenance is needed.

  1. Line 205: why are HEK293T cells described although they are not used in the experiments.

Response: HEK293T cells were only applied in the process of the transfection. After we get the stable transfected cells, the HEK293T cells were not needed for further experiments.

  1. Line 212: replace “cytosol” by “cytosolic”

Response: It has been revised as suggested.

  1. Line 224: “Quantity” has to be replaced by “Quantitative”

Response: It has been revised as suggested.

  1. Line 228 Please correct “Relative Expression of target mRNAs was determined…”

Response: It has been revised as suggested accordingly.

  1. Lines 221-223: “PCR for AKR1C3 promoter detection.” The PCR with these primers seems to be not promoter detection, because the primers are both positioned in the cds. In addition, both primers are given as reverse primers. Please check.

Response: We apologize for the mistake here and it has been corrected in the manuscript.

  1. Chapter 2.14: It has to be made sure that GAPDH the right reference for mRNA quantification, i.e., not itself regulated with drug treatment.

Response: Thank you very much for your suggestion. However, we used GAPDH as reference for PCR experiments in NRF2 knock down cells. This experiment was not treated with drugs. In all, there is no experiment treated with drug that was carried out by PCR in our work.

  1. Line 241: Replace “the” by “a regulation of the”

Response: It has been revised as suggested.

  1. Line 242: Rephrase the sentence “However, western blot showed … and be more precise in statements.

Response: Thank you for your advice. We rephrased the sentence in the manuscript.

  1. Line 245 add “data were” in front of “not statistically…”.

Response: It has been revised as suggested.

Comment on Results part.

  1. Line 265: I would phrase “…, the best overall silencing efficiency was achieved by shAKR1C3-2…”

Response: It has been revised as suggested overall.

  1. Stick to the name “shAKR1C3-2” instead of shAKR1C3” in the remaining text and figures of the manuscript

Response: We replaced all the “shAKR1C3” with name “shAKR1C3-2” in the complete text, figures and figure legends of the manuscript.

  1. Line 275: It is stated that “In addition, AKR1C3 overexpression induced the opposite results (Fig. 2E and Fig. S1K).” This cannot be read without doubt from the figures. Were densiometric measurements performed to verify the statement?

Response: Actually, these markers not increased so significantly in AKR1C3 overexpressing cells (Fig. 2E and Fig. S1K). These markers slightly increased in AKR1C3 overexpressing cells.  Therefore we rephrase the sentence again.

  1. Line 289: Change “in” to “to”

Response: It has been revised as suggested.

  1. Line 290-292: Rephrase sentence: “To better characterize …” The beginning of the sentence suggest that you already know what you would like to show with your experiments.

Response: Thanks for your advice. We changed the original sentence with a new description: “To investigate the role of AKR1C3 in chemotherapy resistance of EAC, apoptosis assay analyzed with flow cytometry and MTT assay were carried out to determine the effect of AKR1C3 on chemotherapy response in EAC cells in vitro”.

  1. Line 298: add “high levels of” before “AKR1C3”

Response: It has been revised as suggested.

  1. Line 299: Replace “Except that” by “Additionally”

Response: It has been revised as suggested.

  1. Lines 299-301: It I stated that “GSEA analysis from TCGA dataset showed drug metabolism related enzymes is up-regulated in AKR1C3-high group of EAC (Fig. S2E). Please name some of the enzymes. From the Fig. 2SE and 5F this information cannot be extracted.

Response: GSEA analysis from TCGA dataset showed drug metabolism related enzymes were up-regulated in AKR1C3-high group of EAC (Fig. S2E). we have listed the gene names such as Alcohol Dehydrogenase 4 (ADH4), UDP Glucuronosyltransferase Family 1 Member A6 (UGT1A6), Alcohol Dehydrogenase 6 (ADH6).

  1. Line 309: replace “and” by “or”

Response: It has been revised as suggested.

  1. Line 315 remove “that”

Response: It has been revised as suggested.

  1. Line 348: Headline: add “cells” or “cell models” after “EAC”

Response: It has been revised as suggested.

  1. Lines 350-352: Verify the statement for OE33 results. Band intensity Responses cannot be read from the figures without doubt. Especially for p-AKT (T308) it is not clear which band is the one that’s important. Were densiometric measurements performed to verify the statement?

Response: As for the western blot results showed that we have two bands for p-AKT (T308). We added a marker aside and pointed out which one that the lower band was the non-specifically band. We appreciated the valuable advice for clear data presenting in the manuscript.

  1. Lines: 360-362: It is stated that “To further clarify whether GSH is regulated through AKR1C3/AKT axis, EAC cells were treated with AKT inhibitor VIII, the result showed that GSH is decreased (Fig. 5D; Fig. S4D)” Does this experiment really prove that AKR1C3 especially is needed in the AKT pathway?

Response: Thank you very much for your reminder. Our results showed that the phosphorylation of AKT was positively regulated by AKR1C3. The phosphorylation of AKT may also be modulated by many other factors via different molecular mechanisms in the cells. From our point of view, AKR1C3 might be not especially needed in the AKT pathway. Our data showed that the phosphorylation of AKT decreased in AKR1C3 knockdown cells and increased in AKR1C3 overexpressing cells, which indicated that AKR1C3 also could regulate AKT pathway in esophageal adenocarcinoma cells.

Comment on Discussion part:

  1. The first sentence starts with a statement that reads like a conclusion: “Our work demonstrates that AKR1C3 could lead to chemotherapy resistance in EAC.” Thus, I would shift this and the 2 following sentences (inclusive figure) to the conclusion or exchange/merge them with the current conclusion.

Response: Thank you very much for your kind suggestions. We have merged the sentences from discussion into the conclusion part, and we added another description to initiation the discussion part. (as shown below)

“Although advances in combination chemotherapy and/or radiotherapy have pro-longed the overall survival of EAC patients, the high rate of resistance to conventional chemotherapy is still the main obstacle to effective therapy of EAC. AKR1C3, as a key member of AKR1Cs subfamily, has been identified as a potential novel therapeutic target in multiple types of cancers. Recently, AKR1C3 has been reported to be upregulated in many human tumors and identified as a prognostic marker in various cancers, including breast cancer, prostate cancer and colon cancer.”

  1. Lines of argumentation should be more targeted, because now the discussion reads in some parts redundant.

Response: Thank you for the advice, we have revised and adjusted the content from the discussion.

  1. Lines 391-392: would you have characteristic markers for Barrett’s esophagus cells to check the hypothesis by WB or PCR?

Response: Thank you very much for your reminder. However, in the practical, it is very hard for us to acquire samples of Barrett’s esophagus from our biobank since only a quite small number of Barrett’s esophagus patients are included so far. Therefore, we did not carry out the experiment to check the hypothesis by WB or PCR. But we have listed the data from public database for the support of evidence.

  1. Line 412 correct “(NADP)” to “(NADP/H)”

Response: It has been revised as suggested.

  1. Line 444: give full name of PTEN

Response: It has been revised as suggested.

  1. GSEA analyses: Which key enzymes are up-regulated? Give at least some names with function of the enzymes. Figure S2E does not give any valuable information without the details

Response: we have listed the gene names such as Alcohol Dehydrogenase 4 (ADH4), UDP Glucuronosyltransferase Family 1 Member A6 (UGT1A6), Alcohol Dehydrogenase 6 (ADH6) and we hope the update version may provide useful information to understand the GSEA findings.

Comment on Figures part:

Figure 1:

  1. make clear that A) and B) show RNA expression data

Response: It has been revised as suggested.

  1. please add sample numbers for the groups in A) and B)

Response: It has been revised as suggested.

  1. Line 251: add “squamous epithelium“

Response: It has been revised as suggested.

  1. Line 254: TCGA EAC which dataset? are these protein or mRNA expression data?

Response: These are mRNA expression data from TCGA-ESCA was applied for survival analysis and EAC subgroup was extracted.

  1. What means N and T and the numbers in Fig 1C?

Response: We explained the abbreviation, N, adjacent normal tissue; T, tumor tissue; at the same time, we added the number of these 12 pairs of EAC tissues and matched adjacent normal tissues.

  1. The antibody against AKR1C3: is it specific against AKR1C3 and not cross-reacting with other AKR1C isoforms?

Response: The antibody against AKR1C3 is from Monoclonal Mouse IgG1 Clone # 871701 to detects human Aldo-keto Reductase 1C3/AKR1C3 in ELISAs and Western Blot from R&D Systems, MAB7678. It demonstrates convinced quality and reacts specifically (Savchuk I, et al. Endocr Connect. 2017 Aug;6(6):348-359. PMID: 28592511).

Figure 2

  1. Line 283: The sentence “Representative images …” would have to be shifted to C) since a picture for the Migration assays is missing. Anyhow, the picture for SKGT-4 in Fig.2C looks not like a colony forming experiment since the colonies are not nicely distributed. Please check.

Response: We apologize that we deleted the representative images of the migration assays, but we did not consider delete the text corresponding to the figures, now we updated accordingly.

  1. Line 285: commas are missing in the enumeration of genes

Response: Commas has been added in the manuscript. We apologize.

  1. Lines 285-287: are statements conform with the pictures in Fig.2E: Please check

Response: We have checked and changed the sentence. (2E) Western-blot results showed that the expression level of CXCR4, ZEB1 and Snail were down-regulated when AKR1C3 was knocked down in SKGT-4 cells. However, these markers were upregulated slightly when AKR1C3 was overexpressed in OE33 cells.

  1. Line 286: “knockdown” has to be replaced by “knocked down”

Response: It has been revised as suggested.

  1. Figure 2D: at which time points was analysis done? Who is the 100%?

Response: Images of EAC cells were captured immediately after scratching as a reference point and were captured again at the end point as indicated. the initially wounded area and serve as a reference point for standardization for the 100%.

Figure 3

  1. Fig 3A: some labels are incomplete

Response: We apologize for the incomplete information. Now we have completed the labels overall.

  1. Fig3C: is concentration unit for cisplatin correct?

Response: The concentration unit for cisplatin is confirmed as correct.

  1. Line 305: Treatment of the different cells strains with different cisplatin concentrations why?

Response: Different EAC cell lines have different sensitivity to chemotherapeutic drugs, we referred the IC50 value to set up corresponding dose to treat cells to induce the apoptosis.

  1. Line 308: rename “Cell cytotoxicity assay” into “Cell viability assay

Response: It has been revised as suggested.

Figure 4

  1. Fig4A: why is x-axis described by DCFDA” instead of H2DCFDA”?

Response: We have updated the DCFDA with H2DCFDA in the manuscript, thank you for the reminding.

  1. Fig 4B: SD are missing? replicates performed?

Response: Sorry for the misunderstanding. SD values are not missing and replicates also been performed. We set all the control one as “1”, for the fold change to this comparison.

  1. Line 344: please rephrase “(around TSS +1400, indicated by black bar).” because there are more black bars and TSS is not explained.

Response: We have rephrased “(around TSS +1400, indicated by black bar).” And we explained TSS and changed the black bars with the red bar accordingly.

  1. Line 345: add “NRF2” in front of “mRNA…”

Response: It has been revised as suggested.

  1. Make Responses in supplement accordingly.

Response: It has been revised as suggested.

Reviewer 2 Report

Zhou and colleagues have investigated the possible association of AKR1C3 (an isoform of the Aldo-keto reductase family) in chemotherapy resistance in esophageal adenocarcinoma. Herein authors demonstrated that overexpression of AKR1C3 could be possibly associated with chemotherapy resistance and its suppression potentiates the efficacy of chemotherapy drugs viz Cisplatin, Oxaliplatin, 5-fluorouracil, and Paclitaxel by mitigating an antioxidant enzyme GSH activity. Authors have shown that AKR1C3 knockdown with siRNA enhances ROS formation and overexpression mitigates ROS production, which could be associated with the activation of antioxidant enzyme transcription factor Nrf2 activation and prevented Akt phosphorylation thereby induces chemotherapy efficacy to kill cancer cells. AKR1C3 is a know steroidogenic enzyme and reported to also contribute to lipid aldehyde production.  Several studies corroborated that the overexpression of AKR1C3 served as possible diagnostic biomarkers and essential prognostic factors for several cancer types such as liver cancer, squamous cell carcinoma of the head and neck etc. 
Free radicals are known to mediate chemotherapeutic response and increased the anti-inflammatory environment mitigates the therapeutic efficiency of anti-cancer drugs. 
The current research article is well constructed and explicitly written. I have few concerns regarding the AKR1C3 mechanism. It is well known that resistance to chemotherapy of cancer cells is multifactorial and redox imbalance is one of the risk factors. Though Authors demonstrated that AKR1C3 inhibition suppressed antioxidant potential and enhanced apoptosis upon chemotherapy treatment.  However, there is no such evidence provided in this study, whether AKR1C3 regulates Nrf2 expression or vice versa or, it's just artifacts of AKR1C3 inhibition? 
By now, it is evident from the literature that the number of intracellular hydrogen peroxide scavengers such as glutathione reductase, Catalase, peroxiredoxin(s) etc. plays a pivotal role in detoxifying H2O2 and know to be upregulated in cancers cells.  Increased lipid peroxidation due to increased ROS, enhanced the production of lipid aldehyde formation such - 4HNE which thereby form an adduct with GSH  (GSH-HNE). Herein figure 5 authors showed increased ROS production and assayed only GSH level in AKR1C3 knockdown cells. It is hard to make any conclusion whether AKR1C3 downregulation prevented GSH level or it also affected another know potential antioxidant? The authors need to determine that whether GSH downregulation in AKR1C3 KD cells was not compensated with other H2O2 detoxifying enzymes. 
It is evident from the literature that DCFDA is non specifically find in all reactive oxygen species.  GSH know to detoxify H2O2 into water and use NADPH+. Authors need to determine intracellular H2O2 with its specific probes. Authors can use commercially available  Peroxy Orange-1 or Mito PY1 to determine H2O2 (cytosolic and mitochondria respectively). 
The authors showed that AKR1C3 inhibition induces cell apoptosis.  However, herein no such data provided to support that how AKR1C3 inhibition induces apoptosis. However, the authors have shown that AKR1C3 inhibition reduced Akt phosphorylation. The authors need to show some downstream markers of apoptosis. it would be better to show cytochrome C release in the cytosol in the above condition. 

Introduction, authors need to elaborate on why they have chosen AKR1C3 over other isoforms? and its role and significance in cancer pathogenesis. 

Author Response

Dear Reviewer,

We appreciate your nice comments and great guidance.  Please find the response to each comment in detail as below shown.

Comment:

  1. I have few concerns regarding the AKR1C3 mechanism. It is well known that resistance to chemotherapy of cancer cells is multifactorial and redox imbalance is one of the risk factors. Though Authors demonstrated that AKR1C3 inhibition suppressed antioxidant potential and enhanced apoptosis upon chemotherapy treatment.  However, there is no such evidence provided in this study, whether AKR1C3 regulates Nrf2 expression or vice versa or, it's just artifacts of AKR1C3 inhibition?

Response: We appreciate your concern. In this study, our results showed the enrichment of NRF2 in the promoter region of AKR1C3 in SKGT-4 by ChIP assay. In addition, the expression of AKR1C3 in both mRNA and protein level were significantly decreased in NRF2- knockdown cells, which indicated that NRF2 could regulate the expression of AKR1C3 in EAC. This regulation of NRF2 on AKR1C3 has been demonstrated in other types of cancer such as colon cancer, renal cell carcinoma, bladder cancer, and lung adenocarcinoma as well (Ge W, et al. Cancer Cell. 2017 Nov 13;32(5):561-573.e6. PMID: 29033244) (Halim M, et al. J Am Chem Soc. 2008 Oct 29;130(43):14123-8. PMID: 18826220). However, whether AKR1C3 could regulate NRF2 is still unclear in EAC cells. It would be really interesting to prove whether AKR1C3 could have a positive or negative feedback on NRF2. Theoretically, AKR1C3 could regulate NRF2 expression via ROS regulation, which however need further evidence in our future project development.

  1. By now, it is evident from the literature that the number of intracellular hydrogen peroxide scavengers such as glutathione reductase, Catalase, peroxiredoxin(s) etc. plays a pivotal role in detoxifying H2O2 and know to be upregulated in cancers cells. Increased lipid peroxidation due to increased ROS, enhanced the production of lipid aldehyde formation such - 4HNE which thereby form an adduct with GSH (GSH-HNE). Herein figure 5 authors showed increased ROS production and assayed only GSH level in AKR1C3 knockdown cells. It is hard to make any conclusion whether AKR1C3 downregulation prevented GSH level or it also affected another know potential antioxidant? The authors need to determine that whether GSH downregulation in AKR1C3 KD cells was not compensated with other H2O2 detoxifying enzymes.

Response: We thank to the valuable suggestion. We apologize that we may not describe very well in the content. In our study, we found that AKR1C3 could positively regulate GSH level, which is in consistent with the change of intracellular ROS level. In addition, GSEA analysis showed that AKR1C3 was positively correlated with GSH synthesis. Therefore, we suggest that AKR1C3 regulate ROS levels at least partly through GSH regulation. It is possible that AKR1C3 may also play a role in the regulation of other potential antioxidants. However, our study could not give a focus on exploring all the potential antioxidants, but we are highly interested in more associated antioxidants for future perspectives.

  1. It is evident from the literature that DCFDA is nonspecifically find in all reactive oxygen species. GSH know to detoxify H2O2 into water and use NADPH+. Authors need to determine intracellular H2O2 with its specific probes. Authors can use commercially available Peroxy Orange-1 or Mito PY1 to determine H2O2 (cytosolic and mitochondria respectively).

Response: We sincerely appreciated the precious advice. In our study, H2DCFDA was used to detect intracellular ROS level in our study, which is generally used in many studies for the direct measurement of intracellular redox states recently. (DeNicola GM et al. Nature. 2011 Jul 6;475(7354):106-9. PMID:21734707) (Zeineldin M et al. Nat Commun. 2020 Feb 14;11(1):913.PMID: 32060267). Our results showed that the intracellular ROS level increased in AKR1C3 knockdown cells and decreased in AKR1C3 overexpressing cells. We did not pay special attention to the H2O2. Therefore, in our study, we did not use H2O2-specific probes such as Peroxy Orange-1 or MitoPY1 to determine intracellular H2O2. However, your suggestion is indeed very Inspirational, and it also provides some new ideas for our further steps on exploring of redox-homeostasis in EAC cells.

  1. The authors showed that AKR1C3 inhibition induces cell apoptosis. However, herein no such data provided to support that how AKR1C3 inhibition induces apoptosis. However, the authors have shown that AKR1C3 inhibition reduced Akt phosphorylation. The authors need to show some downstream markers of apoptosis. it would be better to show cytochrome C release in the cytosol in the above condition.

Response: We sincerely appreciated the helpful advice. Mitochondria plays an essential role in cell apoptosis. The release of cytochrome c could result in the activation of caspases and eventually cell apoptosis (Kalkavan H, et al. Cell Death Differ. 2018 Jan;25(1):46-55. PMID: 29053143), which is dynamically controlled by anti-apoptotic and pro-apoptotic factors, such as BCL2, BAX, and BAK (Ow YP, et al. Nat Rev Mol Cell Biol. 2008 Jul;9(7):532-42. PMID: 18568041). Actually, we explored the effect of AKR1C3 on BAX and BAK. However, the expression level of BAK or BAX were not affected in AKR1C3 OE or KD cells as compared to the control cells (as shown in the figure attached below). Therefore, we suggest that AKR1C3/AKT may regulate apoptosis through GSH controlling. However, the molecular mechanism in more details needs further investigation. For the cytochrome c release assay, it is a very nice tool to show the apoptosis directly, however, it could not reveal more relative information in this case. We determined the apoptosis by Annexin V staining instead. In all, we appreciated your kind suggestion a lot for the critical thinking.

  1. Introduction, authors need to elaborate on why they have chosen AKR1C3 over other isoforms? and its role and significance in cancer pathogenesis.

Response: Thank you very much for the valuable advice. Actually, we selected AKR1C3 for its relatively higher expression in EAC as compared with other AKR1C isoforms in several public datasets (figures as shown below). Now we added a description of the AKR1C subfamily in introduction to better indicate the research focus.

Reviewer 3 Report

This is a well written manuscript, which identifies some mechanisms of Aldo-keto reductase 1C3 (AKR1C3) to increase cell proliferation, colony formation and migration in in vitro cell assays using different cell lines derived from esophageal adenocarcinomas (EAC). Next, the authors present data on the role of AKR1C3 in the induction of chemotherapy resistance again in in vitro cell culture experiments and finally analyze some of the mechanisms related to this effect (i.e. via AKT and ROS).

The authors furthermore provide an analysis from publicly available data sets to show that AKR1C3 is upregulated in EAC and Barrett´s esophagus. In a small own cohort (n=24) derived from a local biobank the expression data were inconclusive. The methods that have been applied are valid (relying on the overexpression and knock-down of AKR1C3). Overall, the data contribute very well to this field of research and the discussion is well written and covers all important aspects in view of the recent literature.

Minor points:

“anti-correlated” in the abstract should be changed to “inversely correlated”;

"etc." should not be used in the abstract

The meaning of the sentence "Anti-cancer therapies based on oxidative damage through direct intracellular reactive oxygen species (ROS) generation, which is an essential step of the apoptotic process [26]" in line 91/92 is not fully clear.

In this study, we aim to investigate the role of AKR1C3 acting as a novel molecular marker to predict the chemotherapy response in EAC: this sentence at the end of the introduction section (line 99/100) should be changed since it does not represent the main focus of this paper.

Line 300: "drug metabolism related enzymes is are upregulated"

The paper of T. Matsunaga et al., Significance of aldo-keto reductase 1C3 and ATP-binding cassette transporter B1 in gain of irinotecan resistance in colon cancer cells. Chemico-Biological Interactions 332, 109295 (2020) could be added in the discussion section.

Author Response

Dear Reviewer,

We appreciate your advice and comments, please find below the detail response to each comment and the revised manuscript accordingly.

Comment: Minor points:

  1. “anti-correlated” in the abstract should be changed to “inversely correlated”;

Response: It has been revised as suggested.

  1. "etc." should not be used in the abstract

Responses: We sincerely appreciated the precious advice, we updated this point.

  1. The meaning of the sentence "Anti-cancer therapies based on oxidative damage through direct intracellular reactive oxygen species (ROS) generation, which is an essential step of the apoptotic process [26]" in line 91/92 is not fully clear.

Response: We rephrase this sentence: “Anti-cancer drugs can induce apoptosis via direct or indirect intracellular reactive oxygen species (ROS) generation in cancer cells.”

  1. In this study, we aim to investigate the role of AKR1C3 acting as a novel molecular marker to predict the chemotherapy response in EAC: this sentence at the end of the introduction section (line 99/100) should be changed since it does not represent the main focus of this paper.

Response: Thank you very much for the advice. We have been deleted the confused part and added a new description in the manuscript.

  1. Line 300: "drug metabolism related enzymes is are upregulated"

Response:  Sorry for the mistake, it has been revised as suggested in the context.

  1. The paper of T. Matsunaga et al., Significance of aldo-keto reductase 1C3 and ATP-binding cassette transporter B1 in gain of irinotecan resistance in colon cancer cells. Chemico-Biological Interactions 332, 109295 (2020) could be added in the discussion section.

Response: We appreciated the valuable advice, now we have cited the reference that reviewer has recommended.

Round 2

Reviewer 1 Report

The manuscript improved, however, the manuscript still contains several linguistic deficiencies and careless mistakes.

I strongly recommend a native English speaker to revise the manuscript.

In part methods are still imprecisely described.

Referencing of instrumentation and material is still inconsistent.

My concerns, which might not cover all issues, are in detail:

Line 94: exchange “important roles” by “an important role”

Line 107: exchange “interactive performance” by “interconnection”

Line 111: change “database” to “databases”

Line 121: correct “anti-SNAI1”

Line 135: exchange “supported” by “provided”

Line 139: change “Penicillin” to “penicillin”

Line 139 and other locations: giving a reference for material once is sufficient

Line 156: (BMG LABTECH) from which country? same reference in line 190 is not necessary

Line 162: change “under microscopy by x40” to “by microscopy using 40x”

Line 165: add “SKGT-4 or OE33” before “cells”

Line 165 change “reach” to “reached”

Line 163-165: How were bands evaluated, i.e. band intensities quantified and normalized?

Line 167-169: Please change sentence to “After carefully washing with phosphate buffered saline (PBS), the SKGT-4 and OE33 cells were incubated in FBS free medium for 12 h and 24 h, respectively,…”

Line 180: add “after incubation” in front of “HRP-“

Line 185 add “, grown” in front of overnight

Line 187: remove “1x”

Line 189: change “MTT solvent” to “MTT dissolving agent”

Line 191: change “serve” to “served” and “ability” to “viability” and “estimated” to “calculated”

Line 192 change “ratio” to “ratios”

Lines 193 and 209: Change sentences to “In all cases biological triplicates were performed.”

Line 197: “Biolegend”. From which country?

Line 199: “stained with H2DCFDA (Sigma-Aldrich)” which concentration? Company from which Country?

Line 205: remove “1 x”

Line 208: “after the deproteinization step by TCA and neutralization by NaHCO3”: be more precise

Lines 212-213: The sentence ” For expression of AKR1C3, the pLenti-CMV-neo vector was purchased from Genscript (Netherlands), using scramble vector as control.” does not make sense. Rephrase. What exactly is the scrample vector?

Line 220: correct “overexprssing”

Line 224: rephrase “supernatant containing the virus was collected through 0.45μm Syringe filters (VWR, Germany) at 48 h and 72 h.”

Line 225: replace “supernatant” by “filtrate”

Line 228: change “for overexpression or knockdown cells” to “AKR1C3 overexpressing or deficient cells”

Line 237 replace “elution” by “eluted”

Lines 243-249 Quantitative RT-PCR:

Was the method applied to the ChiP samples as well? Then include this information here. And give more details on how primers were chosen and designed, and how measured data were analysed. Where are the AKR1C3-ChIP primers located in the genomic sequence.

Line 247: change “were” to “are”

Line 248 : change “by” to “using” In addition give more details on

Line 259 and Fig 1A): define that squamous epithelium is “normal esophagus”

Line 260: change “level” to “levels”

Line 275 Kaplan-Meier survival analysis mentioned in statistical analysis?

Lane 283: change “respected” to “respective”

Line 294: full names of CXCR4, ZEB-1, and Snail1 have to be given once

Line 293-296: The statements “ Besides, AKR1C3 knockdown cells showed decreased expression of metastatic marker CXCR4 and epithelial-mesenchymal transition (EMT) associated factors ZEB-1 and Snail1 at the protein level. (Fig. 2E; Fig. S1F). However, these markers slightly increased in AKR1C3 overexpressing cells (Fig. 2E and Fig. S1K).” still do not convince me: down-regulation of CXCR4 in AKR1C3 knockdown cells and Snail1 up-regulation in AKR1C3 overexpressing cells is not obviously visible in the pictures of Fig.2E. See also Fig 2 legend.

Lines 309: Remove   “To investigate the role of AKR1C3 in chemotherapy resistance of EAC,”

Line 319: insert “that” before “drug-related”

Lines 318-322: I would rephrase to “Additionally, GSEA analysis from the TCGA dataset showed that drug metabolism-related enzymes were enriched in the AKR1C3-high group of EAC (Fig. S2E) including for example alcohol dehydrogenase 4 (ADH4), UDP glucuronosyltransferase family 1 member A6 (UGT1A6) and alcohol dehydrogenase 6 (ADH6).”

Line 330 and 332 replace “treated” by “treatment”

Line 333: remove “treatment”

Line 341: is “induce compulsively” the right wording?

Lines 346 and 347: remove “of”

Line 346 change “overexpression” to “overexpressing”

Line 348: I would remove “Accordingly,” since this is interpretation.

Line 355: “sites” or “site”?

Line 385 change “is” to “was”

Line 397: change “concentration” to “concentrations”

Line 472: change “And” to “Beside,” and “regulator” to “regulation”

Figures:

Figure 2B Units at y-axis are missing

Figure 2D: unify the y-axes, The one in not in percentage

Fig 4A: rename “NRF2” to “anti-NRF2”

Fig4 and 5: “DCFDA” instead of “H2DCFDA” is still to be found at y-axes

Fig 4B: my earlier questions were: SD are missing? replicates performed? These were answered but I do not understand the answers ( Response: Sorry for the misunderstanding. SD values are not missing and replicates also been performed. We set all the control one as “1”, for the fold change to this comparison.) I do not see any fold changes plotted in the figures. Beside the one unit “[uM]” has to be corrected.

Reviewer 2 Report

The authors have tried to address my concerns and can be considered for publication. 

Author Response

Dear Reviewer,

We appreciate your kind advice and comments.

Sincerely,

Yue Zhao

Department of General, Visceral, Cancer and Transplantation Surgery, University Hospital Cologne, Kerpener Straße 62, 50937 Cologne, Germany. yue.zhao@uk-koeln.de; Tel: +49 221 47830601; Fax: +49 221 47830664.